# Excitatory neurotransmission activates compartmentalized calcium transients in Müller glia without affecting lateral process motility

**Joshua M Tworig, Chandler J Coate, Marla B Feller\***

Department of Molecular and Cell Biology, University of California, Berkeley, Berkeley, United States

**Abstract** Neural activity has been implicated in the motility and outgrowth of glial cell processes throughout the central nervous system. Here, we explore this phenomenon in Müller glia, which are specialized radial astroglia that are the predominant glial type of the vertebrate retina. Müller glia extend fine filopodia-like processes into retinal synaptic layers, in similar fashion to brain astrocytes and radial glia that exhibit perisynaptic processes. Using two-photon volumetric imaging, we found that during the second postnatal week, Müller glial processes were highly dynamic, with rapid extensions and retractions that were mediated by cytoskeletal rearrangements. During this same stage of development, retinal waves led to increases in cytosolic calcium within Müller glial lateral processes and stalks. These regions comprised distinct calcium compartments, distinguished by variable participation in waves, timing, and sensitivity to an M1 muscarinic acetylcholine receptor antagonist. However, we found that motility of lateral processes was unaffected by the presence of pharmacological agents that enhanced or blocked wave-associated calcium transients. Finally, we found that mice lacking normal cholinergic waves in the first postnatal week also exhibited normal Müller glial process morphology. Hence, outgrowth of Müller glial lateral processes into synaptic layers is determined by factors that are independent of neuronal activity.

**\*For correspondence:**
mfeller@berkeley.edu

## Editor's evaluation

This manuscript examines the possibility that Müller glia motility in the developing retina prior to eye opening may be controlled or modulated by retinal waves and neural activity. The paper is beautifully written and clear. The data are of high quality and the presentation of the results and discussion balanced, candid and fair.

## Introduction

Bidirectional signaling between neurons and glia is essential for circuit formation and function throughout the nervous system. Across developmental steps from neurogenesis to circuit maturation, glia monitor their environment and in turn regulate neuronal production, migration, and differentiation, promote synapse turnover, and regulate synaptic function via neurotransmitter uptake and ion buffering (*Allen and Lyons, 2018*). In the vertebrate retina, for example, Müller glia exhibit neurogenic potential (*Bernardos et al., 2007*; *Ji et al., 2017*; *Guimarães et al., 2018*), promote circuit-specific wiring via secretion of synaptogenic molecules (*Koh et al., 2019*), regulate phagocytosis of neuronal debris (*Bejarano-Escobar et al., 2017*), and limit neurotransmitter spillover via transporter activity (*Bringmann et al., 2013*).

**eLife digest** When it comes to studying the nervous system, neurons often steal the limelight; yet, they can only work properly thanks to an ensemble cast of cell types whose roles are only just emerging.

For example, 'glial cells' – their name derives from the Greek word for glue – were once thought to play only a passive, supporting function in nervous tissues. Now, growing evidence shows that they are, in fact, integrated into neural circuits: their activity is influenced by neurons, and, in turn, they help neurons to function properly.

The role of glial cells is becoming clear in the retina, the thin, light-sensitive layer that lines the back of the eye and relays visual information to the brain. There, beautifully intricate Müller glial cells display fine protrusions (or 'processes') that intermingle with synapses, the busy space between neurons where chemical messengers are exchanged. These messengers can act on Müller cells, triggering cascades of molecular events that may influence the structure and function of glia. This is of particular interest during development: as Müller cells mature, they are exposed to chemicals released by more fully formed retinal neurons.

Tworig et al. explored how neuronal messengers can influence the way Müller cells grow their processes. To do so, they tracked mouse retinal glial cells 'live' during development, showing that they were growing fine, highly dynamic processes in a region rich in synapses just as neurons and glia increased their communication. However, using drugs to disrupt this messaging for a short period did not seem to impact how the processes grew. Extending the blockade over a longer timeframe also did not change the way Müller cells developed, with the cells still acquiring their characteristic elaborate process networks. Taken together, these results suggest that the structural maturation of Müller glial cells is not impacted by neuronal signaling, giving a more refined understanding of how glia form in the retina and potentially in the brain.

There is extensive evidence that neuronal signaling influences glial physiology in the adult brain. Individual astroglia extend fine processes that contact thousands of synapses and express an array of neurotransmitter receptors (**Allen and Eroglu, 2017**). This enables glia to rapidly integrate neuronal activity, often involving intracellular calcium mobilization. Neuronal activity-evoked calcium events in astroglia range in size from small membrane-proximal microdomains to global cytosolic events mediated by various transmembrane proteins and calcium sources (**Shigetomi et al., 2016**; **Bindocci et al., 2017**; **Nimmerjahn and Bergles, 2015**). Pharmacological studies suggest compartmentalized and global calcium events are mediated by separate mechanisms and evoke different functional responses within astroglia, which lead to differential effects on neurotransmission (**Di Castro et al., 2011**; **Chen et al., 2020**; **Srinivasan et al., 2015**).

Here, we use the mouse retina as a model to explore mechanisms and a possible function of neuron-glia signaling during development. In the retina, Müller glia are the predominant glial type, tiling the entire retinal space and interacting with every retinal cell type (**Wang et al., 2017**). Similar to Bergmann glia of the cerebellum (**De Zeeuw and Hoogland, 2015**; **Lippman et al., 2008**), Müller glia exhibit a radial structure with a stalk extending from the soma, lateral processes extending from the stalk within synaptic layers, and endfeet contacting neuronal somata, axons, and vasculature. In the adult, this complex morphology enables Müller glia to mediate neurovascular coupling, maintain pan-retinal ion homeostasis, and modulate neuronal signaling in the adult (**Newman, 2003**; **Newman, 2015**; **Reichenbach and Bringmann, 2013**). During both mouse (**Rosa et al., 2015**) and zebrafish (**Zhang et al., 2019**) development, Müller glia undergo calcium transients in response to retinal waves, a term used to describe spontaneous bursts of depolarization that propagate across the retina. The functional relevance of wave-associated calcium signaling in Müller glia is not known.

One potential role for wave-associated calcium-signaling in Müller glia is in modulating outgrowth of lateral processes, which initiates and progresses during the same developmental window as retinal waves (**Wang et al., 2017**). In other brain regions and model systems, neuronal activity-evoked calcium transients lead to morphological changes in astroglial processes. For example, in the hippocampus and somatosensory cortex, perisynaptic astrocytic processes undergo spatially localized, glutamate-evoked calcium transients which are essential for activity-evoked ensheathment of

synapses (*Bernardinelli et al., 2014*). In the cerebellum, Bergmann glial processes are highly motile during synaptogenic periods, and their ensheathment of Purkinje cells is impaired when glial gluta-mate transporters are knocked out (*Lippman et al., 2008*; *Miyazaki et al., 2017*). Radial glia of the *Xenopus* optic tectum exhibit neuronal activity-evoked calcium transients and motility during visual system development (*Sild et al., 2016*). In many of these systems, when glial motility is blocked, synaptic development, function, and plasticity are impaired (*Van Horn and Ruthazer, 2019*), high-lighting the importance of glial dynamics in setting up and maintaining neural circuits.

Here, we combine morphological and calcium imaging, electrophysiology, and pharmacology to characterize Müller glial lateral process outgrowth during retinal waves and to determine the impact of neuronal activity on Müller glial morphology.

## Results

### Müller glial lateral process growth is nonuniform and dynamic across the IPL

Our goal is to determine whether spontaneous activity driven by retinal waves influences the morpho-logical development of Müller glial cells. After their differentiation from retinal progenitor cells, which occurs during the first postnatal week, Müller glia extend processes laterally from their stalk into the inner plexiform layer (IPL) (*MacDonald et al., 2017*). To visualize this process, we used the *Slc1a3*-CreER;mTmG reporter line to sparsely label Müller glia (*Figure 1A*). Lateral processes exhib-ited sublayer-specific outgrowth and distribution, and they reached an adult level of complexity soon after eye opening around postnatal day 14 (P14) (*Figure 1B and D*; see *Supplementary files 1 and 2* for statistical comparisons across sublayer and age, respectively). During the first days of Müller glia differentiation, lateral processes preferentially occupied the borders of the IPL (putative S1/S5), with most outgrowth occurring in the prospective ON half of the IPL (S3–S5) until about P10. By eye opening, after P12, processes arborized throughout the IPL and began to resemble their adult distri-bution, characterized by fewer processes in IPL sublayers S2/S4 than in S1/S3/S5. These observations are consistent with previous findings using the same mouse line and immunohistochemistry in fixed tissue (*Wang et al., 2017*).

During outgrowth from the primary stalks, we observed that Müller glial lateral processes across the IPL were highly motile. To characterize process motility, we conducted volumetric two-photon imaging of sparsely labeled processes at a rate of roughly 1 volume every 2 min for 10-min epochs. We characterized motility that occurred during the imaging period by grouping events into distinct categories as follows: new processes that branched from stalks or from existing processes, extension of existing processes, retraction of processes without elimination, elimination of existing processes, or stable for processes that did not change length. We made several observations based on this analysis (*Figure 1E*). First, Müller glial cell lateral processes were highly motile across the IPL throughout the entire second postnatal week. Second, there was a slight bias toward new process sprouting rather than process loss. Third, we found there were roughly equal proportions of extending and retracting processes over 10-min epochs of imaging, highlighting their rapid turnover during this period of development. Finally, we observed a sharp developmental transition toward stability a few days after eye opening (*Figure 1C, E and F*; *Figure 1—figure supplement 1*; *Supplementary file 3*).

To assure that we could observe and quantify changes in process motility, we applied two manipula-tions. First, we slowed process motility with pharmacological agents that disrupt cytoskeletal proteins. Process motility was reduced by bath application of the microtubule-disrupting agent nocodazole (10 µM) and the actin polymerization inhibitor cytochalasin-D (5 µM), the combination of which led to processes stabilization (*Figure 1G*; *Supplementary file 4*). It is unlikely that this loss of motility resulted from damaged cell health, as Müller glial process morphology remained normal in the presence of cytochalasin and nocodazole. Müller glia, like astroglia throughout the brain, exhibit reactive gliosis associated with hypertrophy and expression of filamentous membrane processes when exposed to damage (*Graca et al., 2018*), which we did not observe here. Second, we enhanced process motility by bath application of epidermal growth factor (EGF; one unit [100 ng]/ml; *Figure 1H*; *Supplemen-tary file 5*), which activates EGF receptors (EGFRs). EGFRs are expressed by Müller glia during the second postnatal week (*Close et al., 2006*) and increase motility in other cells via activation of Rac- and Rho-GTPase-dependent pathways (*Pena et al., 2018*; *Liu and Neufeld, 2004*). Taken together,

  

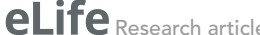

**Figure 1.** Müller glial lateral processes are highly motile during development. (**A**) Orthogonal projections of two-photon volumetric images of Müller glia expressing membrane-GFP (mGFP, green) in sparsely recombined *Slc1a3*-CreER;mTmG retinas at P7 (left) and P47 (right). Cells without Cre-mediated recombination express membrane-tdTomato (magenta). Scale bar: 10 µm. IPL: inner plexiform layer. (**B**) Orthogonal projections of mGFP-expressing Müller glia from P8 to P23. Scale bar: 10 µm. (**C**) Temporally color-coded projections of two-photon Z-stack time series showing motile processes at P12 (top) and stable processes at P23 (bottom). The location and dynamics of lateral processes at each time point can be determined by referencing the color time scale at bottom. Images on right are enlarged insets from images on left to highlight individual processes. Left scale bar: 10 µm; right scale bar: 5 µm. See *Figure 1—figure supplement 1* for enlarged images of these cells in sublayers S1, S3, and S5. (**D**) Distribution of lateral processes across IPL sublayers through development. See *Supplementary file 1* for summary statistics and goodness-of-fit test statistics, and

*Figure 1 continued on next page*

*Figure 1 continued*

*Supplementary file 2* for tests for independent proportions across age groups. (**E**) Proportion of lateral processes in each sublayer that underwent extension (blue), sprouting (light blue), retraction (red), elimination (light red), a combination (purple), or that were stable (gray). (**F**) Proportion of total stable processes across development. See *Supplementary file 3* for statistical comparisons. (**G**) Proportion of total processes that were motile (left) and stable (right) in the absence and presence of blockers of cytoskeletal turnover at P11/12. cyto-D: cytochalasin-D (5 µM); noco: nocodazole (10 µM). See *Supplementary file 4* for statistical comparisons. (**H**) Proportion of total processes that were motile (left) and stable (right) in the absence and presence of epidermal growth factor (EGF, 1 unit [100 ng]/ml) at P8 and P17. See *Supplementary file 5* for statistical comparisons. * corresponds to p < 0.05 for all comparisons. Source data available in *Figure 1—source data 1* and *Figure 1—source data 2*.

The online version of this article includes the following source data and figure supplement(s) for figure 1:

**Source data 1.** Proportions and counts of motile and stable lateral processes across sublayer and age.

**Source data 2.** Proportions and counts of motile and stable lateral processes in the presence and absence of cytoskeletal blockers or EGF.

**Figure supplement 1.** Temporally color-coded projections of the cells shown in *Figure 1C*, highlighting motile and stable processes in sublayers S5 (top), S3 (middle), and S1 (bottom).

these results indicate that lateral process motility is facilitated by turnover of actin filaments and microtubules and may be modulated by growth factor signaling, pathways that potentially interact with neuronal activity (*Georgiou et al., 1999*; *Lavoie-Cardinal et al., 2020*).

## Retinal waves activate compartmentalized calcium transients in Müller glia

Much of the morphological development of Müller glia occurs when retinal waves are present (*Figure 1* and *Wang et al., 2017*). Furthermore, previous studies have demonstrated that retinal waves induce increases in intracellular calcium in Müller glia (*Rosa et al., 2015*; *Zhang et al., 2019*). However, these studies did not assess whether there are distinct calcium compartments within Müller glia that could potentially have distinct impacts on lateral process motility. The two compartments in Müller glia we focused on were their central stalks and their lateral processes within the IPL.

We conducted simultaneous two-photon calcium imaging of Müller glial stalks and processes and voltage-clamp recordings from retinal ganglion cells (RGCs). Several strategies for calcium imaging from Müller glia were used. First, retinas were isolated from *Slc1a3*-CreER; cyto-GCaMP6f$^{flox}$ mice, which express genetically encoded calcium indicator selectively in Müller glia. Second, a subset of retinas were isolated from WT mice and bath-loaded with the chemical calcium dye Cal520, which selectively labels Müller glia (*Rosa et al., 2015*; *Uckermann et al., 2004*). Using these two approaches, we were able to clearly identify the boundaries of Müller glial stalks, while calcium in lateral processes was assessed using regions of interest (ROIs) within the areas of IPL intervening glial stalks (*Figure 2A*). Third, retinas were isolated from mice expressing a membrane-bound calcium reporter in Müller glia (Lck-GCaMP6f in *Slc1a3*-CreER mice) (*Srinivasan et al., 2016*), which enabled resolution of lateral processes but poor detection of cytosolic calcium transients within stalks (*Figure 2—figure supplement 1*). Using all of these approaches, we verified that Müller glial stalks and lateral processes exhibited calcium transients in response to retinal waves as previously reported (*Rosa et al., 2015*), as well as spontaneous, non-wave-associated calcium transients.

We observed several differences in wave-associated calcium transients between stalks and lateral processes. Wave-associated stalk transients propagated throughout the stalk and involved many processes, which we refer to as a global transient. Lateral processes participated in some global transients but also exhibited wave-associated calcium transients independent of stalks (*Figure 2A and B* and *Figure 2—figure supplement 1*). Hence, lateral processes and stalks differentially participated in waves, with a greater proportion of lateral process ROIs responding to waves than stalk ROIs. In addition, wave-associated calcium transients in lateral processes were consistently observed from P9 to P12, while those in stalks were significantly reduced in terms of ΔF/F amplitude and proportion of stalks participating by P11/12 (*Figures 2C and 3A*; *Supplementary file 6*). Finally, the subset of stalks that responded to retinal waves did so with a slight delay relative to lateral processes (*Figure 2B and D*; *Supplementary file 7*). Hence, Müller glial stalks and lateral processes can undergo spatiotemporally distinct calcium signaling events.

We next sought to test whether wave-associated calcium transients in stalks versus lateral processes are mediated via activation of distinct signaling pathways. Muscarinic acetylcholine receptors (mAChRs)

**Figure 2.** Retinal wave-associated calcium transients are compartmentalized within Müller glial stalks and lateral processes. (**A**) *Left,* diagram of experimental setup; cytosolic GCaMP6f was expressed in Müller glia (green) via *Slc1a3*-CreER, and a two-photon microscope was used to image calcium transients in glial stalks and lateral processes in the IPL (dashed line indicates focal depth in whole-mount retina). Excitatory currents were recorded from a retinal ganglion cell (RGC) to detect retinal waves. *Right,* average projection of a field of view (FOV) showing GCaMP6f-expressing Müller glia in retinal whole mount. Upper left panel is the full FOV; upper right panel is a magnified portion of this FOV overlaid with regions of interest (ROIs) for stalks (blue) and lateral processes (yellow); bottom panels show the extent of Müller glial activation by two retinal waves (active regions overlaid in magenta). (**B**) Example traces of fractional change in fluorescence of GCaMP6f-expressing Müller glia showing wave-associated calcium transients in stalks (blue) and processes (yellow) and wave-associated EPSCs recorded from an RGC (black). *Right,* Müller glial calcium response during a retinal wave, expanded in time to highlight temporal delay of stalk transients relative to lateral process transients. Black trace is a voltage-clamp recording from an RGC held at –60 mV. (**C**) Proportion of total stalk and lateral process ROIs that responded to retinal waves at P9/10 and P11/12. Gray dashed line indicates the proportion of ROIs undergoing spontaneous calcium transients at randomly selected times. See *Supplementary file 6* for summary statistics. (**D**) Intercompartment response latency, calculated as latency of stalk responses from the median lateral process response time for retinal waves at P9/10 and P11/12. See *Supplementary file 7* for summary statistics. Image scale bars: 10 μm. Source data available in *Figure 2—source data 1*.

The online version of this article includes the following source data and figure supplement(s) for figure 2:

**Source data 1.** Proportion of total ROIs responding per wave and intercompartment latency among wave-associated calcium transients.

**Figure supplement 1.** Visualization of calcium compartments using *Slc1a3*-CreER;Lck-GCaMP6f[flox].

have been implicated in Müller glial responses to retinal waves (***Rosa et al., 2015***), and isolated Müller glia undergo M1 mAChR-dependent responses to cholinergic agonists (***Da Silva et al., 2008***). Bath application of pirenzepine (5 μM), a selective antagonist of M1 mAChRs, led to a significant reduction in wave-associated calcium transients in stalks and lateral processes (***Figure 3A***). The magnitude of

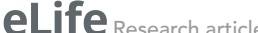

**Figure 3.** M1 mAChRs mediate retinal wave-associated calcium transients preferentially localized to Müller glial stalks. (**A**) Example traces of fractional change in fluorescence of GCaMP6f-expressing Müller glia (top) showing wave-associated calcium transients in stalks (blue) and processes (yellow), and wave-associated EPSCs recorded from a retinal ganglion cell (RGC, bottom) in the absence and presence of pirenzepine (5 µM) at P9 (left) and P11 (right). (**B**) *Left,* cumulative distribution of proportion of stalks and lateral processes that exhibited wave-associated calcium transients in absence (solid line) and presence of pirenzepine (dotted line). *Middle,* box plots showing proportion of ROIs that exhibited wave-associated calcium transients. Gray dashed line indicates the proportion of ROIs that exhibited spontaneous calcium transients at randomly selected times. See ***Supplementary file 8*** for summary statistics. *Right,* fold-change in proportion of ROIs that exhibited wave-associated calcium transients in presence of pirenzepine. See ***Supplementary file 9*** for summary statistics. (**C**) *Left,* cumulative distribution of the z scored ΔF/F amplitude of wave-associated calcium transients in the absence and presence of pirenzepine. See ***Supplementary file 10*** for summary statistics. *Middle,* box plots showing z scored ΔF/F amplitudes separated by age. *Right,* fold-change in wave-associated ΔF/F amplitude in pirenzepine. See ***Supplementary file 11*** for summary statistics. (**D**) Intercompartment response latency for each age and condition. See ***Supplementary file 12*** for summary statistics. (**E**) Summary data for magnitude of compound excitatory post-synaptic currents (EPSC; top) and interwave interval (IWI; bottom) during retinal waves. See ***Supplementary file 13*** for summary statistics. Source data available in ***Figure 3—source data 1***.

The online version of this article includes the following source data and figure supplement(s) for figure 3:

**Source data 1.** Retinal wave response properties among Müller glia and recorded RGCs in ACSF and pirenzepine.

**Source data 2.** Retinal wave response properties among Müller glia in ACSF and DNQX.

**Figure supplement 1.** DNQX reduces glial participation in waves among stalks and lateral processes.

this effect was age- and compartment-dependent, as wave-associated transients in Müller glial stalks had greater sensitivity to M1 mAChR block at P9/10 than at P11/12, and pirenzepine sensitivity was significantly greater in stalks than in lateral processes at early ages (*Figure 3B and C*; *Supplementary files 8–11*). Interestingly, we occasionally detected wave-associated stalk calcium transients that were insensitive to pirenzepine. Pirenzepine-insensitive stalk responses exhibited similar amplitude and latency to those of lateral processes (*Figure 3C and D*; *Supplementary file 12*). These data suggest that wave-associated responses in stalks are controlled by multiple mechanisms in addition to activation of M1 mAChRs. Note, pirenzepine had minimal impact on the amplitude and frequency of compound excitatory postsynaptic currents (EPSCs) and inter-wave interval (IWI, *Figure 3E*; *Supplementary file 13*), indicating pirenzepine had a selective effect on Müller glial M1 mAChRs, rather than on wave-generating circuits.

To confirm that M1 mAChR signaling preferentially impacts wave-associated stalk transients, we enhanced acetylcholine (ACh) release during waves by bath-applying the GABA$_A$ receptor antagonist gabazine (5 µM) (*Wang et al., 2007*). As expected, the presence of gabazine increased the amplitude of EPSCs recorded from RGCs during retinal waves (*Figure 4E*; *Supplementary file 19*). Gabazine also increased the proportion of stalks and lateral process ROIs that underwent wave-associated calcium transients (*Figure 4*; *Supplementary file 14*), but with differential compartment-specific effects: there was a significantly larger gabazine-induced fold-change in proportion of stalks compared with lateral processes responding to waves, and gabazine led to a nearly twofold increase in the amplitude of wave-evoked fluorescence change in stalks, while only slightly increasing the response amplitude in lateral processes (*Figure 4C*; *Supplementary files 15–17*). Subsequent application of pirenzepine reduced the extent, magnitude, and latency of stalk responses to waves while largely sparing lateral process responses (*Figure 4B–D*; *Supplementary files 14–18*) and without altering wave properties (*Figure 4E*; *Supplementary file 19*). This effect was independent of age from P9 to P12; pirenzepine suppressed wave-associated transients in stalks following gabazine-mediated upregulation of responses throughout this period. Taken together, these data indicate that M1 mAChRs mediate wave-associated global calcium transients in Müller glial stalks and support the conclusion that stalks and lateral processes are functioning as distinct calcium compartments, with lateral processes undergoing non-M1 mAChR-mediated calcium transients during retinal waves.

## Müller glial process motility occurs independent of wave-associated calcium transients

Is Müller glial lateral process motility impacted by neuronal activity? To test whether neuronal activity influences Müller glial lateral process motility, we assessed the impact of multiple pharmacological agents on process motility after the first postnatal week (*Figure 5*). First, we assessed the impact of agents that modulated mAChR signaling resulting from neural activity (*Figure 5A*). Gabazine, which potentiated calcium transients in Müller glial processes and stalks, in part via activation of M1 mAChRs, did not change the proportion of processes undergoing motility. Similarly, there was no significant change in lateral process motility in the presence of pirenzepine, which blocks M1 mAChRs and preferentially abolished wave-associated calcium transients in Müller glial stalks (*Figure 5A and F*; see *Supplementary file 20* for summary statistics). Second, we assessed the impact of glutamatergic signaling, which is also implicated in wave-associated Müller glial calcium transients (*Rosa et al., 2015*; *Zhang et al., 2019*). TBOA blocks glutamate transporters and enhances wave-associated calcium transients in Müller glia by increasing glutamate spillover, while DNQX and AP5 block glutamatergic retinal waves and reduce calcium transients in Müller glia (*Rosa et al., 2015*; *Blankenship et al., 2009*; *Figure 3—figure supplement 1*). Similar to our results in gabazine and pirenzepine, the presence of TBOA or DNQX/AP5 had no significant impact on motility (*Figure 5B and F*). These observations together indicate that lateral process motility does not arise from wave-associated calcium transients in stalks or processes.

To further explore a role for ACh and glutamate in influencing motility, we tested whether the local release of neurotransmitter would alter the motility of nearby processes, as is the case for perisynaptic astrocytic processes elsewhere in the brain, where it is postulated that local elevations in neurotransmitter promote process growth and synapse coverage (*Bernardinelli et al., 2014*; *Arizono et al., 2020*). We locally perfused via continuous iontophoresis the AChR agonist carbachol or glutamate onto Müller glial processes in *Slc1a3*-CreER;mTmG retinas with sparse Cre-mediated recombination.



**Figure 4.** GABA$_A$ receptor block potentiates wave responses in stalks via M1 mAChR activation. (**A**) Example traces of fractional change in fluorescence of GCaMP6f-expressing Müller glia showing wave-associated calcium responses (top) and wave-associated excitatory post-synaptic currents (EPSCs) recorded from an RGC (bottom) in control (left), 5 µM gabazine (middle), and gabazine plus 5 µM pirenzepine (right) at P11. (**B**) *Left*, cumulative distributions and inset box plots of proportion of stalk ROIs (top) and lateral process ROIs (bottom) that exhibited wave-associated calcium transients in control (solid line, A), gabazine (dotted line, G), and gabazine/ pirenzepine (dashed line, G/P) from P9 to P12. Gray dashed line in box plots indicates the proportion of ROIs undergoing spontaneous calcium transients at randomly selected times. *Right,* fold-change (FC) in proportion ROIs responding in gabazine versus ACSF (top) and in pirenzepine vs. gabazine (bottom). Gray dashed line indicates FC of 1. See

*Figure 4 continued*

***Supplementary files 14 and 15*** for summary statistics. (**C**) *Left,* cumulative distributions of normalized wave-associated calcium transient amplitudes in ACSF, gabazine, and gabazine/pirenzepine from P9 to P12, with inset box plots generated from the same data. *Right,* FC in normalized calcium response amplitude after application of gabazine (top) and then pirenzepine (bottom). See ***Supplementary files 16 and 17*** for summary statistics. (**D**) Intercompartment latency in wave-associated calcium transients in control, gabazine (gbz), and gabazine/pirenzepine (gbz+pir), from P9 to P12. See ***Supplementary file 18*** for summary statistics. (**E**) Summary data for magnitude of (top) and interval between (bottom) compound EPSCs associated with retinal waves in control, gabazine, and gabazine/pirenzepine for paired FOVs from P9 to P12. See ***Supplementary file 19*** for summary statistics. Source data available in ***Figure 4—source data 1***. FOV, field of view; RGC, retinal ganglion cell; ROIs, regions of interest.

The online version of this article includes the following source data for figure 4:

**Source data 1.** Retinal wave response properties among Müller glia and recorded RGCs in ACSF, gabazine, and gabazine plus pirenzepine.

In parallel experiments using retinas from *Slc1a3*-CreER;GCaMP6f^flox mice, we confirmed that this method induced strong calcium transients in stalks and lateral processes. Despite this, we observed no impact of local perfusion of agonists on Müller glial lateral process motility (***Figure 5C, D and F***).

As a final test for whether intracellular calcium signaling in Müller glia is required for motility, we assessed motility after bath-loading the potent calcium chelator, BAPTA-AM. Despite blocking retinal waves and all calcium transients in Müller glia (***Figure 5—figure supplement 1***), the presence of BAPTA did not alter the proportion of processes exhibiting motility. Taken together these data suggest that motility of Müller glial lateral processes persists in the absence of intracellular calcium transients or neuronal activity (***Figure 5E and F***). We note however that some of these observations were made using relatively small numbers of cells, and we cannot rule out a small effect of these pharmacological perturbations given our sample size and high cell-to-cell variability in lateral process motility.

## Long-term blockade of cholinergic retinal waves does not alter Müller glial morphology

Thus far, we have assessed the role of neuronal activity and calcium signaling on Müller glial lateral process motility using live imaging during the second postnatal week, while there is lateral process outgrowth. However, during the first postnatal week, there are wave-associated calcium transients in Müller glia also mediated by activation of mAChRs (***Rosa et al., 2015***). To test whether retinal wave-associated calcium signaling during this first postnatal week influences later process outgrowth during the second postnatal week, we assessed Müller glial morphology in retinas isolated from mice lacking the β2 nicotinic acetylcholine receptor subunit (β2-nAChR-KO). β2-nAChR-KO mice exhibit significantly reduced cholinergic retinal waves (***Bansal et al., 2000***; ***Burbridge et al., 2014***; ***McLaughlin et al., 2003***) and therefore have reduced M1 mAChR-induced signaling in glial cells.

Morphology of individual Müller glial cells was assessed by filling them with Alexa-488 via sharp pipette in P12 and P30 wild-type (WT) and β2-nAChR-KO mice. Filled cells were visualized via two-photon volumetric imaging in live tissue, and processes were traced for subsequent morphological analysis (***Figure 6A–C***). At both P12 and P30, individual dye-filled Müller cells exhibited lateral processes throughout the IPL, with more processes in the ON-half of the IPL than in the OFF-half prior to eye opening, in agreement with ***Figure 1***. Detailed assessment shows that by P12, Müller glial lateral processes exhibited nearly the same level of complexity as observed in the adult. In addition, we found no significant difference between WT control and β2-nAChR-KO Müller glia in terms of several morphological parameters including complexity, number, area, and length of lateral processes. However, we did observe a significant reduction in number of primary branches projecting from the stalk in β2-nAChR-KO Müller glia before eye opening, while in mature retina all morphological properties of in β2-nAChR-KO Müller glia appeared normal (***Figure 6D and E***; ***Supplementary file 21***). This suggests that although retinal waves may play a role in process sprouting from Müller glial stalks during development, other mechanisms produce lateral processes in normal numbers and appearance as the retina matures.



**Figure 5.** Müller glial lateral process motility is unaffected by manipulations of retinal waves. (**A**) Temporally color-coded projections of two-photon Z stack time series showing motile and stable processes in control (ACSF), gabazine (5 µM), and gabazine plus pirenzepine (5 µM) at P11. Scale bar: 10 µm. (**B**) Same as A in control (ACSF), TBOA (25 µM), and TBOA plus DNQX (20 µM) and AP5 (50 µM) as pharmacological manipulations of retinal waves.

*Figure 5 continued on next page*

*Figure 5 continued*

(**C** and **D**) show motility (left, middle) and calcium responses (right, scale bar: 20 μm) to iontophoretically applied carbachol (P10) and glutamate (P9). A sharp electrode (indicated in magenta) was filled with 10 mM agonist and current continuously applied to eject agonist into the IPL near mGFP-expressing processes to image motility, or near GCaMP6f-expressing processes to image calcium responses. (**E**) *Left,* motility persisted at P12 even when intracellular calcium was chelated via 200 μM BAPTA-AM bath application, which greatly reduced baseline calcium ($F_0$) and eliminated neuronal activity (see *Figure 5—figure supplement 1*). Scale bars: 10 μm. *Right,* average projections (top) and normalized traces of Lck-GCaMP6f fluorescence (bottom) in DMSO and BAPTA. (**F**) Summary data showing proportion of total processes exhibiting stability (top) and other categories of motility (bottom) for drug conditions shown in (**A–E**). * gabazine was included in the control condition for a subset of experiments using pirenzepine, as in (**A**). † TBOA was included in the control condition for a subset of experiments using DNQX/AP5, as in (**B**). These trials are denoted by dotted lines connecting control and drug conditions. See Figure *Supplementary file 20* for summary statistics. Source data available in *Figure 5—source data 1*.

The online version of this article includes the following source data and figure supplement(s) for figure 5:

**Source data 1.** Proportions and counts of motile and stable lateral processes during perturbations of neuronal activity or intracellular calcium.

**Figure supplement 1.** Bath loading with BAPTA-AM during development eliminates glial and neuronal calcium activity and retinal waves.

## Discussion

In this study, we showed that as Müller glial lateral processes emerge during the second postnatal week of development, they exhibit both rapid motility and retinal wave-associated calcium transients that are partially compartmentalized. Global calcium transients in stalks were delayed relative to transients in lateral processes, were inhibited by an M1 mAChR antagonist, and became smaller and less frequent after P10. In contrast, wave-associated calcium transients in lateral processes often occurred locally and independent of stalk transients, were less sensitive to M1 mAChR block, and persisted throughout the second postnatal week. However, in contrast to astrocytes and radial glia in other regions of the central nervous system (*Chen et al., 2020*; *Lippman et al., 2008*; *Bernardinelli et al., 2014*; *Sild et al., 2016*), Müller glial lateral process motility and outgrowth occurred independent of neuronal signaling and calcium transients. These results indicate that Müller glial process morphology in the IPL is not regulated by neuronal activity.

### Mechanisms underlying Müller glial process motility

Glial morphology is critical for normal development of circuits. For example, a plexus of glial processes provides a diffusional barrier for neurotransmitters and other signaling molecules (*Bringmann et al., 2013*; *Syková, 2001*), undergoes contact-mediated signaling with neurites to modulate synapse formation and function (*Filosa et al., 2009*; *Murai et al., 2003*), and participates in synaptic pruning via activation of phagocytic pathways in glia (*Chung et al., 2013*).

Across brain regions, neuron-glia signaling impacts glial morphology in diverse ways. We found in the retina that Müller glial lateral process motility during development was not impacted by neuron-glia signaling via release of neurotransmitter. This is similar to Bergmann glia, radial glia of the cerebellum, whose process motility is developmentally regulated and not directly affected by perturbations of neuronal activity or calcium influx (*Lippman et al., 2008*). Similarly in microglia, another highly motile cell type, chelation of calcium with BAPTA-AM slightly slowed but did not block process motility (*Pozner et al., 2015*). In striatal astrocytes, FRET-based synaptic proximity assays have shown that although fine branches interact with synapses, these interactions are stable and unaffected by neuronal activity (*Octeau et al., 2018*). These findings stand in contrast to the perisynaptic processes of hippocampal astrocytes, in which metabotropic glutamate receptor (mGluR) activation leads to localized increase in intracellular calcium which promotes process motility and coverage of dendritic spines (*Bernardinelli et al., 2014*). This divergence between Müller glia and hippocampal astrocytes might be reflective of differences in ultrastructural interactions between astroglial processes and synapses in these two systems: perisynaptic processes of hippocampal astrocytes engage in true tripartite synapses in a brain region with high plasticity (*Ventura and Harris, 1999*; *Haber et al., 2006*), while there is currently no published evidence revealing similar structures in the retina. Further, astrocytic processes express specific actin-binding proteins such as ezrin which enable coupling of

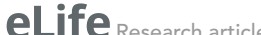

**Figure 6.** Chronic perturbation of retinal waves does not impact distribution, length, or complexity of Müller glial lateral processes. (**A**) Orthogonal projections of two-photon Z stacks obtained after sharp-filling single Müller glial cells with Alexa-488 at P12 (top) and at P30 (bottom) in wild-type (WT, left) and in β2 nicotinic acetylcholine receptor knockout retinas (β2KO, right). Image scale bar is 10 µm. (**B**) XZ projections of skeletons from cells shown in (**A**). (**C**) XY projections of skeletons from cells shown in (**A**). (**D**) Morphological measurements obtained from traced and skeletonized Müller glia. See **Supplementary file 21** for summary statistics. Source data available in **Figure 6—source data 1**. (**E**) Sholl intersection profiles for WT (black) and β2KO (red) at P12 (top) and P30-39 (bottom). Left plots are Sholl profiles carried out on XY projections, middle plots are from XZ projections, and right plots

*Figure 6 continued on next page*

*Figure 6 continued*

are from YZ projections. For XZ and YZ projections, the center of the Sholl radius was placed on portion of the stalk closest to the inner nuclear layer (INL). Two-way mixed ANOVA revealed no significant effect of genotype on number of intersections at any Sholl radii for both ages. GCL, ganglion cell layer. Source data for Sholl analyses available in *Figure 6—source data 2*.

The online version of this article includes the following source data for figure 6:

**Source data 1.** Comparisons of morphological properties between wild type and β2-nAChR-KO Müller glia.

**Source data 2.** Sholl analysis profiles for wild type and β2-nAChR-KO Müller glial lateral processes.

cytoskeletal dynamics with signaling through mGluRs (*Lavialle et al., 2011*), which were previously shown to have minimal contribution to retinal wave-associated responses in Müller glia (*Rosa et al., 2015*).

Multiple pathways independent of neuronal activity may regulate morphological dynamics in Müller glial processes. One possibility is the release of growth factors from neurons or other nearby cells, which has been implicated in morphological changes among astrocytes in other systems (*Wu et al., 2017*; *Liu and Neufeld, 2007*). This hypothesis is supported by our finding that application of exogenous EGF enhanced process motility prior to eye opening. Interestingly, EGFR expression in Müller glia is high prior to eye opening and declines with a developmental time course that matches our observed decline in motility (*Close et al., 2006*). Another intriguing possibility is that repulsive homotypic interactions between lateral processes from neighboring Müller glia underly their motility. This idea is supported by single-cell photoablation experiments in zebrafish Müller glia and mouse Schwann cells, in which processes from non-ablated cells filled in the territory vacated by ablated cells (*Williams et al., 2010*; *Brill et al., 2011*). Further study will be required to determine whether these or other alternative pathways mediate Müller glial lateral process motility during development.

## Calcium compartments in Müller glia

We observed two distinct calcium compartments in Müller glia activated by retinal waves. Compartmentalization of calcium within Müller glial stalks was achieved via activation of glial M1 mAChRs, while lateral processes exhibited non-M1 mAChR-mediated transients with a smaller latency than those in stalks. We observed an age-dependent reduction in stalk participation in waves, while lateral processes continued to respond through P12. (Note, our observation that lateral processes continue to respond to retinal waves from P9 to P12 contrasts with a previous study that reported an overall reduction in Müller glial responses to waves at these ages [*Rosa et al., 2015*]. We attribute this difference to improved sensitivity of GCaMP6f over GCaMP3). The reduced stalk participation observed in older animals may be due to downregulation of M1 mAChRs prior to eye opening, or to a reduction in volume release of acetylcholine during the transition to glutamatergic waves (*Ford et al., 2012*). Our observation that gabazine application during glutamatergic waves caused wave-associated stalk transients to return in an M1 mAChR-dependent manner suggests that the latter is true.

Gabazine application also enhanced wave-associated calcium transients in lateral processes via a mechanism independent of M1 mAChR activation. We suspect this is due to enhanced release of other excitatory neurotransmitters following relief of GABA$_A$ receptor-mediated inhibition in the presence of gabazine during retinal waves. This is similar to the action of TBOA, which blocks glutamate transporters and subsequently increases retinal wave frequency and wave-associated Müller glial calcium transients due to an increase in extracellular glutamate levels (*Rosa et al., 2015*).

Because astroglia and neurons both express neurotransmitter receptors and transporters, it is interesting to consider which of these cell populations is targeted by the antagonists used in this study. Current RNA-sequencing datasets suggest that Müller glia do not express GABA receptors at an appreciable level and therefore it is unlikely that gabazine directly impacts signaling in these cells. That said, Müller glia express GABA transporters which might contribute to signaling during retinal waves (*Bringmann et al., 2013*; *Macosko et al., 2015*). They also express glutamate transporters (EAAT1) which are targeted by TBOA. At 25 µM, the concentration of TBOA used to modulate retinal waves in this study was likely too low to effectively inhibit glial EAAT1 (IC$_{50}$: 70 µM), and more likely affects retinal wave properties via neuronal EAATs (*Jabaudon et al., 1999*). Furthermore, although we did not test whether pirenzepine acts on neuronal mAChRs, it is possible that blockade of non-Müller glial M1 mAChRs by pirenzepine indirectly affects glial responses to retinal waves, as previous studies

have revealed expression of these receptors in retinal interneurons (**Macosko et al., 2015**; **Strang et al., 2010**; **Strang et al., 2015**). However, our observation that retinal wave-associated EPSC amplitude and IWI remained normal in the presence of pirenzepine is consistent with the interpretation that pirenzepine acts directly on glial M1 mAChRs rather than on wave-generating circuits (**Figures 3E and 4E**).

## Roles of calcium compartmentalization in astroglia

Subcellular calcium compartmentalization plays a variety of roles in neuron-glia signaling and tissue homeostasis (**Wang et al., 1997**). Hippocampal astrocytic processes undergo at least two types of spatiotemporally distinct calcium transients in response to nearby synaptic activity, depending on the type of synaptic activity that occurs. This calcium signaling is thought to regulate synaptic transmission within the astrocytic territory (**Di Castro et al., 2011**). Similarly, compartmentalized calcium transients during startle response frequently occur in distal branches, and less frequently in somata of hippocampal astrocytes in awake mice. Distinct signaling pathways, including those involving transmembrane calcium channels or transporters, adrenergic receptors, and IP3 receptors, differentially contribute to calcium transients in branches versus somata (**Denizot et al., 2019**). Another study used a computational approach to reveal how calcium influx in fine astrocytic branches can be mediated by glutamate transporter-dependent activity of $Na^+/Ca^{++}$ exchangers, while somatic calcium can be modulated by mGluR-dependent activation of $IP_3$ signaling (**Oschmann et al., 2017**). Compartmentalization between Müller glial stalks and processes may arise by a similar mechanism.

We found that wave-associated compartmentalized calcium activity is not required for Müller glial process motility, so this activity likely plays a role in other functions of Müller glia during retinal development. These functions include calcium-dependent neurovascular coupling (**Biesecker et al., 2016**), release of gliotransmitters such as ATP or D-serine (**Newman, 2003**; **Newman, 2015**; **Sullivan and Miller, 2010**), or secretion of synaptogenic molecules such as thrombospondin or growth factors (**Koh et al., 2019**; **de Melo Reis et al., 2008**). Our identification of M1 mAChR as a driver of wave-associated calcium transients in stalks provides a target for selective perturbation in Müller glia to better define a role for these transients in retinal circuit development.

# Materials and methods

## Key resources table

| Reagent type (species) or resource | Designation | Source or reference | Identifiers | Additional information |
|---|---|---|---|---|
| Strain, strain background (*Mus musculus*; male & female) | C57BL/6J (wild-type) | The Jackson Laboratory | RRID:IMSR_JAX:000664 | Background strain used for mouse crosses |
| Strain, strain background (*M. musculus*; male and female) | *Slc1a3*-CreER; Tg(Slc1a3-cre/ERT)1Nat/J; also known as GLAST-CreER | The Jackson Laboratory | RRID:IMSR_JAX:012586 | Expresses tamoxifen-inducible Cre/ERT in Müller glia |
| Strain, strain background (*M. musculus*; male and female) | mTmG; mT/mG; B6.129(Cg)-Gt(ROSA)26Sor$^{tm4(ACTB-tdTomato,-EGFP)Luo}$/J | The Jackson Laboratory | RRID:IMSR_JAX:007676 | Used to measure morphology/motility in Müller glia |
| Strain, strain background (*M. musculus*; male and female) | GCaMP6f$^{flox}$; B6.129S(Cg)-Gt(ROSA)26Sor$^{tm95.1(CAG-GCaMP6f)Hze}$/J | The Jackson Laboratory | RRID:IMSR_JAX:024105 | Used to measure calcium dynamics in Müller glial stalks and processes |
| Strain, strain background (*M. musculus*; male and female) | Lck-GCaMP6f$^{flox}$; C57BL/6N-Gt(ROSA)26Sor$^{tm1(CAG-GCaMP6f)Khakh}$/J | The Jackson Laboratory | RRID:IMSR_JAX:029626 | Used to measure calcium dynamics in Müller glial processes |
| Strain, strain background (*M. musculus*; male and female) | PDGFRa-Cre; C57BL/6-Tg(Pdgfra-cre)1Clc/J | The Jackson Laboratory | RRID:IMSR_JAX:013148 | Used to verify loss of calcium transients following BAPTA-AM application |

*Continued on next page*

*Continued*

| Reagent type (species) or resource | Designation | Source or reference | Identifiers | Additional information |
|---|---|---|---|---|
| Strain, strain background (*M. musculus*; male and female) | β2-nAChR-KO; C57BL/6J-*Chrnb2^{tm1Mdb}* | *Xu et al., 1999*; Bansal et al., *J Neurosci*, 2000. doi: https://doi.org/10.1523/JNEUROSCI.20-20-07672.2000 | RRID:MGI:2663180 | Targeted deletion generated by Xu et al. (*J Neurosci*, 1999) and maintained in our lab since |
| Chemical compound, drug | 4-hydroxytamoxifen (4-OHT, 50:50 E:Z isomers) | Sigma-Aldrich | T176 | Injected at variable doses to induce uniform or sparse Cre-mediated recombination in Müller glia |
| Chemical compound, drug | Cytochalasin-D (from *Zygosporium mansoni*) | Avantor/VWR | Supplier no.: 250255 catalog number: 80055-214 | Disrupts actin filaments and inhibits actin polymerization |
| Chemical compound, drug | Nocodazole | Sigma-Aldrich | M1404 | Disrupts microtubule assembly/disassembly |
| Chemical compound, drug | Epidermal growth factor (EGF) (Rac/Cdc42 Activator II) | Cytoskeleton, Inc | CN02 | EGF receptor agonist |
| Chemical compound, drug | Pirenzepine dihydrochloride | Tocris | 1071 | M1 mAChR antagonist |
| Chemical compound, drug | Gabazine (SR 95531 hydrobromide) | Tocris | 1262 | GABA_A receptor antagonist |
| Chemical compound, drug | DL-TBOA | Tocris | 1223 | Glutamate transporter (EAAT) antagonist |
| Chemical compound, drug | DNQX disodium salt | Tocris | 2312 | AMPAR antagonist |
| Chemical compound, drug | DL-AP5 | Tocris | 3693 | NMDAR antagonist |
| Chemical compound, drug | BAPTA-AM | Tocris | 2787 | Intracellular calcium chelator |
| Chemical compound, drug | Carbachol (carbamoylcholine chloride) | Tocris | 2810 | Cholinergic agonist |
| Chemical compound, drug | L-glutamic acid (Glu) | Sigma-Aldrich | 56-86-0 | Glutamatergic agonist |
| Chemical compound, drug | Alexa fluor 488/594 hydrazide, sodium salt (Alexa-488/594) | Thermo Fisher Scientific | A10436/A10438 | Used for filling Müller glia to visualize processes, and as a counterdye for focal agonist application |
| Software, algorithm | ScanImage | Vidrio Technologies, LLC | RRID:SCR_014307 | Two-photon image acquisition software |
| Software, algorithm | FIJI/ImageJ | Created at National Institutes of Health; Schindelin et al., *Nature Methods*, 2012. doi:10.1038/nmeth.2019 | RRID:SCR_002285 | Used for image processing and analysis |

*Continued on next page*

*Continued*

| Reagent type (species) or resource | Designation | Source or reference | Identifiers | Additional information |
|---|---|---|---|---|
| Software, algorithm | MATLAB | MathWorks | RRID:SCR_001622 | Custom scripts used for data processing, analysis |
| Software, algorithm | NormCorre: Non-Rigid Motion Correction | Flatiron Institute, Simons Foundation; Pnevmatikakis and Giovannucci, *J Neurosci Methods*, 2017. doi: https://doi.org/10.1016/j.jneumeth.2017.07.031 | | For 2D registration of calcium imaging movies |
| Software, algorithm | FIJI plugin: Distance Transform Watershed | Legland et al., *Bioinformatics*, 2016. doi: 10.1093/bioinformatics/btw413 | | For segmentation of Müller glial stalks in calcium imaging movies |
| Software, algorithm | FIJI plugin: Correct 3D Drift | Parslow et al., *J Vis Exp*, 2014. doi: 10.3791/51086. | | For 3D registration of glial morphology time courses |
| Software, algorithm | FIJI plugin: Simple Neurite Tracer | Arshadi et al., *Nat Methods*, 2021. doi:10.1038/s41592-021-01105-7 | RRID:SCR_016566 | For tracing of dye-filled glial processes and stalks |
| Software, algorithm | R; RStudio | R Project for Statistical Computing; RStudio | RRID:SCR_001905; RRID:SCR_000432 | Used for statistical analyses |

## Animals

All mice were purchased from The Jackson Laboratory and were maintained on mixed C57BL/6 backgrounds. For motility experiments, P8-P116 *Slc1a3*-CreER;mTmG mice were generated by cross breeding *Slc1a3*-CreER mice (strain 012586; also known as GLAST-CreER) with *Rosa26$^{mTmG}$* mice (strain 007676) (**Wang et al., 2017**). *Slc1a3*-CreER mice express tamoxifen-inducible Cre recombinase under control of a Müller glia-specific promoter. mTmG is a dual-fluorescence reporter line that constitutively expresses membrane-bound tdTomato, and upon Cre-mediated recombination expresses membrane-bound green fluorescent protein (mGFP). For calcium imaging experiments, P9-P12 *Slc1a3*-CreER;GCaMP6f$^{flox}$ mice were generated by crossing *Slc1a3*-CreER mice with GCaMP6f$^{flox}$ (cytosolic) mice (strain 024105) or Lck-GCaMP6f$^{flox}$ (membrane-bound) (**Srinivasan et al., 2016**) mice (strain 029626) to enable specific and inducible calcium indicator expression in Müller glia. To verify loss of calcium transients in neurons and Müller glia following BAPTA-AM application, we include imaging results from *PDGFRα*;cyto-GCaMP6f$^{flox}$ mice (*PDGFRα*-Cre: strain 013148), which label Müller glia and a subset of neurons with cytosolic GCaMP6f, as well as results from *Slc1a3*-CreER;Lck-GCaMP6f$^{flox}$ retinas (*Figure 5—figure supplement 1*). For sharp fills of Müller glial cells with fluorescent dye, we used C57BL/6 mice as controls for comparison with mice lacking the β2 subunit of the nicotinic acetylcholine receptor (β2-nAChR-KO).

Cre-mediated recombination was induced via intraperitoneal injection of 4-hydroxytamoxifen (50:50 E and Z isomers, Sigma-Aldrich) dissolved in sunflower seed oil. For uniform expression across the retina in neonates, injections of 0.5 mg tamoxifen were made 2 and 4 days before each experiment. For sparse expression to enable resolution of individual Müller glia during imaging, a single injection of 0.1 mg tamoxifen was made 2 days before each experiment.

All animal procedures were approved by the University of California, Berkeley Animal Care and Use Committee and conformed to the NIH Guide for the Care and Use of Laboratory Animals, the Public Health Service Policy, and the SFN Policy on the Use of Animals in Neuroscience Research.

## Retinal preparation

For all experiments, male and female mice were anesthetized via isoflurane inhalation and decapitated. Eyes were enucleated and retinas dissected in oxygenated (95% $O_2$/5% $CO_2$) artificial cerebrospinal fluid (ACSF) at room temperature under bright field (less than P10) or infrared (P10 and above) illumination. ACSF contained (in mM) 119.0 NaCl, 26.2 NaHCO3, 11 glucose, 2.5 KCl, 1.0 K2HPO4,

2.5 CaCl2, and 1.3 MgCl2. Isolated retinas were mounted ganglion cell side up on filter paper (Millipore) and transferred into the recording chamber of an upright microscope for imaging and electrophysiological recording. Retinas were continuously superfused with oxygenated ACSF (2–4 ml/min) at 32–34°C for the duration of experiments and kept in the dark at room temperature in oxygenated ACSF when not imaging or recording. In a subset of experiments, WT retinas were bath-loaded with the organic calcium dye Cal520 (12 µM) for 1.5–2 hr prior to performing calcium imaging.

## Two-photon imaging

Two-photon imaging of Müller glia in the IPL was performed using a modified movable objective microscope (MOM; Sutter Instruments) equipped with an Olympus 60×, 1.0 NA, LUMPlanFLN objective (Olympus America). Two-photon excitation was evoked with an ultrafast pulsed laser (Chameleon Ultra II; Coherent) tuned to 920 nm for all fluorophores. The microscope was controlled by ScanImage software (http://www.scanimage.org). Scan parameters were (pixels/line×lines/frame [frame rate in Hz]): 256×256 (1.48–2.98), at 1–2 ms/line. When imaging GCaMP6f fluorescence in Müller glial stalks and processes, the focal plane was set to ~1/3 the distance from the ganglion cell layer to the inner nuclear layer. Volumetric imaging of motility in mGFP-expressing Müller glial processes and of surrounding tdTomato-expressing cells was achieved by acquiring sequential two-channel Z-stacks through the entire IPL, with slices 1 µm apart and averaging four frames per slice. Volumes were taken every 2 min for a total of 10 min of imaging lateral process dynamics. A similar procedure was used for imaging motility in a subset of cells after filling them with fluorescent dye.

## Electrophysiological recordings

Whole-cell voltage-clamp recordings were made from whole-mount retinas while simultaneously imaging GCaMP6f fluorescence. Under infrared illumination, RGC somas were targeted for voltage-clamp recordings using glass microelectrodes with resistance of 3–5 MΩ (PC-10 pipette puller; Narishige) filled with an internal solution containing (in mM) 110 CsMeSO$_4$, 2.8 NaCl, 20 HEPES, 4 EGTA, 5 TEA-Cl, 4 Mg-ATP, 0.3 Na$_3$GTP, 10 Na$_2$Phosphocreatine, and QX-Cl (pH 7.2 and 290 mOsm). The liquid junction potential correction for this solution was –10 mV. Signals were acquired using pCLAMP10 recording software and a MultiClamp 700A amplifier (Molecular Devices), sampled at 20 kHz and low-pass filtered at 2 kHz.

## Pharmacology

For pharmacology experiments, after 5–10 min of recording data in ACSF, pharmacological agents were added to the perfusion, and experimental recordings were obtained 5 min afterward. Drug concentrations were as follows: 5 µM cytochalasin-D (Avantor), 10 µM nocodazole (Sigma-Aldrich), 1 unit (100 ng)/ml EGF (Cytoskeleton, Inc), 5 µM pirenzepine (Tocris), 5 µM gabazine (Tocris), 25 µM DL-TBOA (Tocris), 20 µM DNQX (Tocris), 50 µM AP5 (Tocris), and 200 µM BAPTA-AM (Tocris). DL-TBOA and BAPTA-AM were prepared in 0.1% DMSO. For calcium chelation experiments, whole-mount retinas were incubated in BAPTA-AM or vehicle (ACSF/0.1% DMSO) for 1.5–2 hr, and then moved to ACSF for 30 min prior to imaging (*Shigetomi et al., 2008*). To verify that BAPTA-AM loading abolished retinal waves and residual calcium activity in cytosolic and membrane-proximal compartments, we measured calcium activity in neurons and Müller glia using cyto-GCaMP6f$^{flox}$ (crossed with *PDGFRα*-Cre) or Lck-GCaMP6f$^{flox}$ (crossed with *Slc1a3*-CreER) at P10–P12 before and after BAPTA-AM loading.

## Focal agonist iontophoresis

For focal agonist application to mGFP-expressing lateral processes in *Slc1a3*-CreER;mTmG retinas, carbachol (10 mM, Tocris), or glutamic acid (10 mM, Sigma-Aldrich) was loaded into sharp electrodes pulled on a P-97 Micropipette Puller (Sutter) with a resistance of 100–150 MΩ. Electrodes also contained 2 mM Alexa-594 for verification of iontophoresis and to visualize electrodes under two-photon illumination. While imaging at 920 nm and visualizing fluorescence from mGFP and Alexa-594, electrodes were driven into the IPL at a ~15° angle using a micromanipulator set to the 'DIAG' function. For each cell, the tip of the electrode was placed 1–5 µm from the arbor of lateral processes of interest. Iontophoresis of agonist or control was achieved by applying continuous current using pCLAMP10 software. To apply carbachol, which is positively charged at physiological pH, continuous

current of +4 nA was used, while –4 nA current was used to apply glutamate, which is negatively charged. Current of opposite polarity to that used for each respective agonist application was applied as a control, and the order of control versus agonist application was shuffled to negate potential effects resulting from damage caused by the electrode. In a separate set of experiments, iontophoresis of agonist was performed in *Slc1a3*-CreER;cyto-GCaMP6f^flox or *Slc1a3*-CreER;Lck-GCaMP6f^flox retinas to verify that this method reliably evoked calcium transients in lateral processes.

## Müller glial sharp fills

For a subset of experiments testing the role of retinal wave-evoked calcium transients in Müller glial motility (*Figure 5*), and to compare Müller glial morphology in WT and β2-nAChR-KO retinas (*Figure 6*), sharp electrodes were pulled as described above, and the tip was bent 15–20° using a microforge. Electrodes were loaded with 2 mM Alexa-488 in water, and iontophoresis of dye into single Müller glial cells was achieved by applying a –10 nA pulse for 500 ms in MultiClamp 700A software while the electrode was on the membrane of Müller glial endfeet in the ganglion cell layer. Electrodes were withdrawn as soon as cells started to fill with dye, and cells were imaged 1–5 min after filling (*Ding et al., 2016*). Two-photon volumetric images of dye-filled lateral processes of single Müller glial cells in the IPL were acquired using the same image parameters used for imaging motility, as described above.

## Image processing and analysis: calcium imaging

All images were processed using custom scripts in FIJI/ImageJ (National Institutes of Health) (*Schindelin et al., 2012*) and MATLAB (MathWorks). For calcium imaging movies, following non-rigid motion correction (*Pnevmatikakis and Giovannucci, 2017*), ROIs were semi-automatically placed over stalks using a filtering algorithm based on a Laplace operator and segmented by applying a user-defined threshold (*Dorostkar et al., 2010*). The Distance Transform Watershed function (*Legland et al., 2016*) in FIJI was used to segregate nearby stalks in binarized images. This method defined most of the ROIs that an experienced user would recognize by eye. Manual adjustments were made to include stalks that were missed and to remove ROIs that were erroneously placed on structures other than stalks, which were defined as semiregularly spaced, punctate regions of fluorescence 2–3 µm in diameter in average intensity projection images. Lateral process ROIs were subsequently defined by randomly placing 250 2.5×2.5 µm² squares in regions that did not overlap with stalk ROIs. Fluorescence intensity is reported as the average intensity across all pixels within the area of each ROI, and normalized as the relative change in fluorescence (ΔF/F) as follows:

$$\frac{\Delta F}{F} = \left( F - F_0 \right) / F_0,$$

where F is the instantaneous fluorescence at any time point and $F_0$ is the baseline fluorescence, defined as the median fluorescence value over the duration of the trace.

ΔF/F traces were smoothed using a two-frame median filter and analyzed to detect calcium transients using custom MATLAB code. Traces were Z scored, and a threshold Z score of 3 was used to define calcium transients, which were then further defined as wave-associated if they occurred within 3 s of the peak of a wave-associated EPSC. Intercompartment latency was defined as the difference, in seconds, between wave-associated transients in stalks and the median wave-associated transient time for all lateral processes within the FOV, for each retinal wave.

## Image processing and analysis: Müller glial morphology

For volumetric images of Müller glial process morphology, images were bandpass filtered in XY space to reduce noise while maintaining the structure of fine processes, and registered using the Correct 3D Drift plugin in FIJI (*Parslow et al., 2014*). 10-minute volumetric time series were corrected for photobleaching using the 'Histogram matching' setting within FIJI's Bleach Correction plugin. Time series were collapsed into temporally color-coded images to facilitate identification of motile and stable processes in 3D. The Cell Counter tool within FIJI was used to count lateral processes and define their morphological dynamics as one of the following: extending, retracting, new process, lost process, extension followed by retraction, retraction followed by extension, and stable. Locations of process tips within the IPL were recorded and binned into one of five equally sized sections, corresponding to

putative sublayers S1 through S5, defined by their distance from the INL and GCL borders as identified using membrane-tdTomato fluorescence on somas.

Volumetric images of sharp-filled Müller glia were traced using the Simple Neurite Tracer plugin (FIJI) (*Arshadi et al., 2021*). We traced each glial stalk and subsequently any visible processes branching from the stalk, creating distinct paths for each process and preserving branch order relationships. Measurements including number of tips, total process length, total branch points, and primary branches from stalk were derived from traced paths. For Sholl and convex hull analyses, paths were converted to binary skeletons and registered to correct for XY displacement of the stalk between the GCL and INL. Sholl analysis was performed using concentric rings spaced 1 µm apart (*Ferreira et al., 2014*). For the XY plane, the center of each Sholl radius was placed on the registered stalk, and intersections were counted at each radius overlayed on a maximum projection image. For the XZ and YZ planes, the center of each Sholl radius was placed on the end of the stalk closest to the INL in orthogonal projections of traced cells. Sholl radii were normalized to the total IPL thickness. Convex hull area was defined as the area of the smallest convex polygon enclosing the entire skeleton in an XY maximum projection image.

## Statistical analysis

Group measurements are expressed as mean ± standard error of mean (SEM) except when nonparametric test results are reported, in which case median, 1st, and 3rd quartiles are reported. MATLAB and RStudio were used to carry out all statistical tests. Chi-squared tests of goodness-of-fit were used to test for nonuniformity in lateral process distribution across the IPL within each age. Chi-squared tests for independent proportions were used to test for age-dependent differences in process distribution across the IPL (*Supplementary files 1 and 2*). When comparing the proportion of total stable processes between control and experimental manipulations, we applied Wilcoxon signed-rank tests when cells were paired between conditions, and Wilcoxon rank-sum tests for unpaired cells. When there were less than five cells in a particular condition, we pooled process counts between cells and reported chi-square test statistics for these comparisons as well (*Supplementary files 3–5 and 20*). To test for age-, compartment-, and drug-dependent differences in retinal wave-associated calcium transients (*Supplementary files 6; 8–11; 14–17*), we applied two- or three-way mixed ANOVAs followed by post hoc t-tests with Benjamini-Hochberg correction for multiple comparisons, when appropriate. Paired t-tests were used to compare fold-changes between compartments in response to pirenzepine and gabazine (*Supplementary files 9; 11; 15 and 17*). Intercompartment latency of wave-associated calcium transients was compared between ages and conditions using Wilcoxon rank-sum tests, and within each condition tested for significant difference from zero using Wilcoxon signed-rank tests (*Supplementary files 7; 12 and 18*). Wave EPSC amplitude and IWI were compared between control and drug conditions using paired t-tests (*Supplementary files 13 and 19*). Sholl intersection profiles between WT and β2-nAChR-KO Müller glia were compared using two-way repeated-measures mixed ANOVA to test for genotype-associated differences in complexity across Sholl radii (*Figure 6*). Two-sample t-tests were used for comparison of other morphological measurements between WT and β2-nAChR-KO Müller glia (*Supplementary file 21*).

## Acknowledgements

The authors thank members of the Feller lab for commenting on the manuscript. JMT was supported by the National Science Foundation Graduate Research Fellowship (DGE 1752814). JMT and MBF were supported by NIH grants R01EY019498, R01EY013528, and P30EY003176.

## Additional information

### Competing interests

Marla B Feller: Reviewing editor, eLife. The other authors declare that no competing interests exist.

## Funding

| Funder | Grant reference number | Author |
|---|---|---|
| National Science Foundation | DGE 1752814 | Joshua M Tworig |
| National Institutes of Health | R01EY019498 | Joshua M Tworig<br>Marla B Feller |
| National Institutes of Health | R01EY013528 | Joshua M Tworig<br>Marla B Feller |
| National Eye Institute | P30EY003176 | Joshua M Tworig<br>Marla B Feller |

The funders had no role in study design, data collection and interpretation, or the decision to submit the work for publication.

## Author contributions

Joshua M Tworig, Conceptualization, Data curation, Formal analysis, Investigation, Methodology, Validation, Visualization, Writing – original draft, Writing – review and editing; Chandler J Coate, Data curation, Investigation; Marla B Feller, Conceptualization, Funding acquisition, Project administration, Supervision, Writing – original draft, Writing – review and editing

## Author ORCIDs

Joshua M Tworig  http://orcid.org/0000-0001-7798-4480
Marla B Feller  http://orcid.org/0000-0002-9137-5849

## Ethics

This study was performed in strict accordance with the recommendations in the Guide for the Care and Use of Laboratory Animals of the National Institutes of Health. All of the animals were handled according to approved institutional animal care and use committee (IACUC) protocols of the University of California, Berkeley. The protocol was approved by the University of California Animal Care and Use Committee Office for Animal Care and Use (Protocol Number: AUP-2015-10-8080-1).

## Decision letter and Author response

Decision letter https://doi.org/10.7554/eLife.73202.sa1
Author response https://doi.org/10.7554/eLife.73202.sa2

# Additional files

## Supplementary files

• Transparent reporting form

• Supplementary file 1. Analysis of differences in lateral process outgrowth between sublayers; separate tests for each age group.

• Supplementary file 2. Analysis of age-dependent differences in lateral process outgrowth within each sublayer.

• Supplementary file 3. Analysis of age-dependent differences in proportion stable processes.

• Supplementary file 4. Analysis of motility in blockers of cytoskeletal rearrangements.

• Supplementary file 5. Analysis of motility in exogenous EGF.

• Supplementary file 6. Analysis of calcium compartmentalization: proportion of ROIs participating in waves.

• Supplementary file 7. Analysis of intercompartment latency in ACSF.

• Supplementary file 8. Proportion of total ROIs participating in retinal waves: ACSF versus pirenzepine.

• Supplementary file 9. Fold-change in proportion ROIs participating in retinal waves: ACSF versus pirenzepine.

• Supplementary file 10. Z scored wave-associated ΔF/F amplitude: ACSF versus pirenzepine.

• Supplementary file 11. Fold-change in wave-associated ΔF/F amplitude: ACSF versus pirenzepine.

- Supplementary file 12. Intercompartment latency in ACSF versus pirenzepine.
- Supplementary file 13. Wave properties in ACSF versus pirenzepine.
- Supplementary file 14. Proportion ROIs participating in retinal waves in ACSF, gabazine, and gabazine + pirenzepine.
- Supplementary file 15. Fold-change in proportion ROIs participating in waves in ACSF versus gabazine, and in gabazine versus gabazine + pirenzepine.
- Supplementary file 16. Z scored wave-associated ΔF/F amplitude in ACSF, gabazine, and gabazine + pirenzepine.
- Supplementary file 17. Fold-change in wave-associated transient amplitude: ACSF versus gabazine; gabazine versus gabazine+ pirenzepine.
- Supplementary file 18. Intercompartment latency in ACSF, gabazine, and gabazine + pirenzepine.
- Supplementary file 19. Wave properties in ACSF, gabazine, and gabazine + pirenzepine.
- Supplementary file 20. Comparisons of proportion stable processes across conditions.
- Supplementary file 21. Comparisons of morphological properties between wild-type (WT) and β2-nAChR-KO Müller glia.

### Data availability
Source data for all figures is available and is uploaded to a linked Dryad repository as well as directly with this submission.

The following dataset was generated:

| Author(s) | Year | Dataset title | Dataset URL | Database and Identifier |
|---|---|---|---|---|
| Tworig J, Feller M, Coate C | 2021 | Data from: Excitatory neurotransmission activates compartmentalized calcium transients in Müller glia without affecting lateral process motility | https://doi.org/10.5061/dryad.6078/d12d9f | Dryad Digital Repository, 10.5061/dryad.6078/d12d9f |

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
