## [Editor Report]

This manuscript examines the possibility that Müller glia motility in the developing retina prior to eye opening may be controlled or modulated by retinal waves and neural activity. The paper is beautifully written and clear. The data are of high quality and the presentation of the results and discussion balanced, candid and fair.

---

## [Decision Letter]

**Decision letter after peer review:**

Thank you for submitting your article "Excitatory neurotransmission activates compartmentalized calcium transients in Müller glia without affecting lateral process motility" for consideration by *eLife*. Your article has been reviewed by 3 peer reviewers, and the evaluation has been overseen by Gary Westbrook as the Senior and Reviewing Editor. The following individual involved in review of your submission has agreed to reveal their identity: Edward S Ruthazer (Reviewer #2). The reviewers have discussed their reviews with one another, and the Senior Editor has drafted this letter to help you prepare a revised submission.

Summary

The reviewers were in agreement that this is a high quality manuscript that makes a substantial contribution to the field. There are a few essential points that we think will either require additional data or a more balanced discussion of the limitations of the experiments.

Essential revisions:

1. There was concern that the observations in Figure 5 were underpowered. The statistics used are well described but the n is small. Please either increase the sample size or explicitly state the limitations on interpretations imposed by the small sample.

2. Due diligence in the BAPTA-AM experiments to rule out any Ca transients remaining in tiny processes, e.g. by using Lck-GCaMP. We understand that it is very hard to rule out residual Ca activity w/o membrane-localized GCaMP. Please either provide that evidence or tone down conclusions from that experiment.

*Reviewer #1 (Recommendations for the authors):*

The paper by Tworig et al., reports high quality data from careful experiments aimed at addressing an important question related to how Muller glial (astrocyte-like cells) move during development and during neuronal activity. The authors have used developmental biology and physiology of the retina to great effect to explore in a systematic way the questions mentioned. The paper is beautifully written and very clear. The data are of high quality and the presentation of the results and discussion balanced, candid and fair.

I am very excited to see such quality work as in this manuscript. I have several suggestions to improve the paper and to make it more impactful for the readers. These could be addressed fairly easily with a round of revisions. Overall, I think this is an excellent paper. Congratulations to the authors on high quality work that addresses an idea that has been lingering in the literature for a long time but one that has never been satisfactorily nailed. I very much look forward to citing this paper when it is published.

1. I would encourage the authors to include an additional figure that shows something similar to Figure 1C, but in a way that is larger. Showing that at a larger scale early on will help the reader more easily follow the paper and importantly allow them to get a sense of the raw data and what was measured.

2. Although the figures are very well constructed and very clear, there are a few areas of improvement. The words Wave 1 and Wave 2 are really hard to see in Figure 2a. In Figure 2, the authors should add the concentration of pirenzepne to the figure panels (also add drug concentrations to all the figure panels). The representative images in Figure 6A,B, C are very nice but too small. I suggest remaking this figure to make these panels larger.

3. The authors performed several pharmacological experiments to explore if neurons affect Muller glia motility and calcium. Do they have data using antagonists of ATP P2Y receptors? If they do, this result could be added. However, I don't think it is critical and so new experiments are strictly not needed, but the result could be added if they have the data at hand. This is relevant as several labs have shown astrocytes and neurons release ATP. I suspect nothing will happen and it would be useful to say that, if indeed that is the case.

4. I think the discussion is very fair and balanced. There is only one previous paper that comes to mind that used a complex FRET assay and also found that astrocyte processes did not move relative to synapses even when astrocyte calcium went up during afferent stimulation. That paper could be usefully cited and does not detract from the current study – in fact supports it. However, this is not essential and it's up to the authors to decide if they want to cite it. If they do choose to cite it, the Pubmed ID is PMID: 29621490 and relevant data are towards the end of the results.

*Reviewer #2 (Recommendations for the authors):*

1. Was there a consistent difference in process to stalk latency with GCaMP vs Cal520? A serious concern associated with the use of any Ca dye but especially with large protein-based genetically encoded Ca indicators is that they may physically hinder the free diffusion of Ca within the cell. Performing this experiment using an organic dye as well as a GECI permits this issue to be addressed, but the paper does not clearly distinguish between latency data obtained with the two imaging methods. Could this be compared and discussed?

2. While the existing data presented are adequately compelling, it might have been interesting to apply a manipulation that altered wave interval rather than intensity to demonstrate unambiguously that the drugs used are acting via neurons rather than through some possible receptor binding on the MG themselves.

3. The demonstration that BAPTA-AM prevented Ca transients is important in MG but less meaningful for neurons where membrane voltage fluctuations could continue to signal in the absence of any intracellular Ca changes. Is the quantification shown mainly for glial Ca or for neuronal Ca? Again, I am willing to accept that if it works on neurons it likely works on glia but what was measured in this case could be made clearer in the text.

4. My main criticism is in the morphometric analysis section where the two major manipulations that might test neuronal activity's influence on glial motility suffered from:

a) being fairly underpowered with N's of 5 and 4. For a negative result that seems to be too few cells.

b) the blockade always came on a background of upregulation (pip on gabazine and APV/DNQX on TBOA). Even so the effects were reported as showing a non-significant trend. Is there any way to either repeat these experiments in the absence of the upregulating compounds or at least to do the statistical comparisons to the baseline in the absence of that upregulating compound. As far as I could tell baselines for these experiments compared upregulation vs upreg+ downreg which would have been an outstanding control if there had been a positive result but which is less convincing with a marginal negative result. Because the main conclusion of the entire paper largely rests on these experiments, I would prefer to see them done properly.

*Reviewer #3 (Recommendations for the authors):*

Overall this is a strong and exciting manuscript investigating important questions about neural interactions by using the retina as a model system. However, some clarifications and controls are needed to ensure that the author's main claim is fully supported by the data provided before publication.

Concerns:

Throughout the text, the word glia is used to depict astrocytes and Muller glia. However, the glia name also includes other cell types such as microglia and oligodendrocytes. Thus the use of glia instead of astroglial cell types may generate confusion. Particularly when generalized statements are made concerning calcium transients.

The manuscript is well-written, clear, and accessible to a broad audience. However, the authors should improve Figures to extend the accessibility. For example, Figure 1A could start with a schematic of the IPL layers in the retina so the significance of the labeling, such as S1 S2 and so on, can be understood by diverse audiences.

Related to Figure 1F and G:

Please clearly describe in the manuscript how the authors ensured that pharmacological treatments that stopped Muller glial process motility did not do so by damaging cell health.

Related to figure 3B:

Pirenzepine treatment affected both stalk and lateral process wave proportions. This effect is not mentioned or discussed in the Results section.

Is M1 mAChR only expressed by Muller glia in the retina? Citing literature investigating this question would benefit the discussion of the impacts of pharmacological manipulations on Muller Glia.

Related to figure 4:

Gabazine has a strong effect on lateral process calcium event frequency, which is M1 mAChR independent. This is not mentioned in the results. But interesting.

Related to figure 5 E and F:

Bath application of BAPTA was used to block all retinal waves and all calcium transients in Mueller glia. However, the evidence provided for the lack of calcium transients in lateral processes under BAPTA treatment is weak. How did the authors ensure that there were no residual calcium transients in the lateral processes of Muller glia in the presence of BAPTA? To rule out any residual calcium, a similar experiment using Lck-GCaMP in Muller glia is needed.

Related to Figure 6:

The authors used a genetic model to impair retinal waves and domesticated its impact on Muller glial morphology. They found no significant differences between the morphological parameters of Muller glia protrusions between the two genotypes at two different ages. While average morphological parameters seem to be similar between the genotypes, the distribution of data points looks different, indicating a role for the waves in helping Muller glia establish similar size balance territories within the synaptic layers. Comparing the distribution of data points between two genotypes can be informative.

Related to all data presented: The number of cells and mice used for each quantitative analysis should be given in the figure legend as well as in the provided statistical tables. The sex of mice used should also be included in the legends for each figure.

---

## [Author Response]

Essential revisions:1. There was concern that the observations in Figure 5 were underpowered. The statistics used are well described but the n is small. Please either increase the sample size or explicitly state the limitations on interpretations imposed by the small sample.

We have conducted additional experiments. However, given the high cell-to-cell variability in process motility, some of the comparisons made in Figure 5 remain underpowered for detecting small effect sizes. We have addressed this limitation in the results (page 10, lines 218-221). Moreover we have added conditions in which motility was measured in pirenzepine or DNQX/AP5 alone, without the prior addition of gabazine or TBOA, to address comment 4b from Reviewer 2. All of this data is included in the revised Figure 5, and the results text has been updated on page 9 and 10 to reflect minor changes in interpretation.

2. Due diligence in the BAPTA-AM experiments to rule out any Ca transients remaining in tiny processes, e.g. by using Lck-GCaMP. We understand that it is very hard to rule out residual Ca activity w/o membrane-localized GCaMP. Please either provide that evidence or tone down conclusions from that experiment.

Indeed, we did verify the loss of cytosolic and membrane-localized calcium activity in Müller glia following BAPTA-AM application using both cytosolic and Lck-GCaMP in separate experiments. This has been clarified in the manuscript (lines 215; 363-367; 423-426). We have also included this data explicitly in a new figure supplement, Figure 5—figure supplement 2.

Reviewer #1 (Recommendations for the authors):The paper by Tworig et al., reports high quality data from careful experiments aimed at addressing an important question related to how Muller glial (astrocyte-like cells) move during development and during neuronal activity. The authors have used developmental biology and physiology of the retina to great effect to explore in a systematic way the questions mentioned. The paper is beautifully written and very clear. The data are of high quality and the presentation of the results and discussion balanced, candid and fair.I am very excited to see such quality work as in this manuscript. I have several suggestions to improve the paper and to make it more impactful for the readers. These could be addressed fairly easily with a round of revisions. Overall, I think this is an excellent paper. Congratulations to the authors on high quality work that addresses an idea that has been lingering in the literature for a long time but one that has never been satisfactorily nailed. I very much look forward to citing this paper when it is published.1. I would encourage the authors to include an additional figure that shows something similar to Figure 1C, but in a way that is larger. Showing that at a larger scale early on will help the reader more easily follow the paper and importantly allow them to get a sense of the raw data and what was measured.

We have included a new figure supplement with larger images from the cells shown in Figure 1C (Figure 1—figure supplement 3).

2. Although the figures are very well constructed and very clear, there are a few areas of improvement. The words Wave 1 and Wave 2 are really hard to see in Figure 2a. In Figure 2, the authors should add the concentration of pirenzepne to the figure panels (also add drug concentrations to all the figure panels). The representative images in Figure 6A,B, C are very nice but too small. I suggest remaking this figure to make these panels larger.

All figure panels were adjusted to increase visibility as needed. Where there was space within figure panels, drug concentrations were included. All drug concentrations can be found in figure legends and in the *Pharmacology* subsection within the Materials and methods section (pages 18-19).

3. The authors performed several pharmacological experiments to explore if neurons affect Muller glia motility and calcium. Do they have data using antagonists of ATP P2Y receptors? If they do, this result could be added. However, I don't think it is critical and so new experiments are strictly not needed, but the result could be added if they have the data at hand. This is relevant as several labs have shown astrocytes and neurons release ATP. I suspect nothing will happen and it would be useful to say that, if indeed that is the case.

We have not performed experiments in the presence of ATP P2Y antagonists for this study. However, a previous study from our lab revealed that the nonselective P2 antagonist suramin did not block wave-associated calcium transients in Muller glia, suggesting that ATP does not contribute to wave-associated responses in these cells during retinal waves (Rosa et al., *eLife*, 2015).

4. I think the discussion is very fair and balanced. There is only one previous paper that comes to mind that used a complex FRET assay and also found that astrocyte processes did not move relative to synapses even when astrocyte calcium went up during afferent stimulation. That paper could be usefully cited and does not detract from the current study – in fact supports it. However, this is not essential and it's up to the authors to decide if they want to cite it. If they do choose to cite it, the Pubmed ID is PMID: 29621490 and relevant data are towards the end of the results.

Thank you for bringing this study to our attention. We have included it in the discussion (lines 269-271) along with a comparison to other glial types.

Reviewer #2 (Recommendations for the authors):1. Was there a consistent difference in process to stalk latency with GCaMP vs Cal520? A serious concern associated with the use of any Ca dye but especially with large protein-based genetically encoded Ca indicators is that they may physically hinder the free diffusion of Ca within the cell. Performing this experiment using an organic dye as well as a GECI permits this issue to be addressed, but the paper does not clearly distinguish between latency data obtained with the two imaging methods. Could this be compared and discussed?

We only used GCaMP6f to measure stalk-process latency for Figures 2 and 3, but for Figure 4 we also included 5 fields of view from Cal520 bath-loaded retinas to assess latency in gabazine and pirenzepine. Direct comparison of latencies between GCaMP6f and Cal520-loaded Muller glia in this subset of experiments revealed a systematically higher latency in Cal520-loaded retinas than in GCaMP6f-expressing retinas. This could be due to a number of factors, including better signal-to-noise with Cal520, more robust loading of stalks with Cal520 versus expression of GCaMP6f in stalks, contamination of lateral process responses with fast neuronal responses due to nonspecific Cal520 loading, or changes in free diffusion of calcium in the presence of Cal520 vs. GCaMP6f. Because we observed a stalk-response latency greater than zero in both Cal520-loaded and GCaMP6f-expressing retinas, our interpretation that stalks and lateral processes operate as distinct calcium compartments is still supported.

We have included the analysis for the benefit of the reviewer (Author response image 1 and Author response table 1). We are not including it in the revision since the basis of this difference is not clear. However, we can add this as an additional supplement to the paper if the reviewers/editor deem it as important.

**Author response table 1. sa2table1:** Comparison of intercompartment latency between GCaMP6f-expressing and Cal520-loaded Muller glia.

Indicator	GCaMP6f	Cal520	
n wave-assoc. stalk transients	ACSF: 41gabazine: 1953	ACSF: 38gabazine: 1630	Pairwise comparison across indicators (Wilcoxon rank-sum)
ACSF latency; signed-rank p-value	0.17 [0.0/0.85]p = 9.0 x 10^-3^	1.35 [1.10/1.69]p = 9.5 x 10^-8^	p = 2.8 x 10^-6^
Gabazine latency; signed-rank p-value	0.34 [0.0/1.01]p = 3.7 x 10^-117^	1.35 [1.01/2.03]p = 2.0 x 10^-259^	p = 8.8 x 10^-267^

**Author response image 1. sa2fig1:** Comparison of intercompartment latency of retinal wave-associated calcium transients between GCaMP6f-expressing and Cal520-loaded Muller glia. Although intercompartment latency was significantly greater than zero using both GCaMP6f and Cal520, Cal520-loaded Muller glia exhibited a significantly higher latency than GCaMP6f- expressing Muller glia during retinal waves. Median [1^st^ quartile/3^rd^ quartile], ACSF (GCaMP6f, Cal520; seconds): 0.169 [0.0/0.85], 1.35 [1.10/1.69]. Gabazine (GCaMP6f, Cal520; seconds): 0.34 [0.0/1.01]; 1.35 [1.01/2.03]. Red bars indicate mean latency for each group. * indicates p < 0.05. Differences in latency between indicators were tested using Wilcoxon rank-sum, and differences in latency from zero were tested using Wilcoxon signed-rank tests.

2. While the existing data presented are adequately compelling, it might have been interesting to apply a manipulation that altered wave interval rather than intensity to demonstrate unambiguously that the drugs used are acting via neurons rather than through some possible receptor binding on the MG themselves.

We have used two manipulations that alter wave frequency – gabazine and TBOA. Current RNA-sequencing datasets suggest that Muller glia do not express GABA receptors at an appreciable level and therefore it is unlikely that gabazine impacts signaling in these cells. That said, GABA could bind the GABA transporters on Muller glia which might contribute to their signaling. TBOA on the other hand can bind to Muller glial glutamate transporters (EAAT1) at high concentrations (IC_50_, EAAT1: 70 μM), but we used 25 μM which is closer to the IC_50_ values for neuronal EAATs (6 μM for EAAT2 and EAAT3). Although we did not measure wave interval directly in the presence of TBOA, TBOA has previously been shown by our lab and others to reduce the inter-wave interval during glutamatergic waves due to an increase in extracellular glutamate levels (Rosa et al., *eLife,* 2015). We have included this discussion in the revised manuscript (page 15).

3. The demonstration that BAPTA-AM prevented Ca transients is important in MG but less meaningful for neurons where membrane voltage fluctuations could continue to signal in the absence of any intracellular Ca changes. Is the quantification shown mainly for glial Ca or for neuronal Ca? Again, I am willing to accept that if it works on neurons it likely works on glia but what was measured in this case could be made clearer in the text.

Indeed, we did verify the loss of cytosolic and membrane-localized calcium activity in Müller glia following BAPTA-AM application using both cytosolic and Lck-GCaMP in separate experiments. This has been clarified in the manuscript (lines 215; 363-367; 423-426). We have also included this data explicitly in a new supplementary figure (Figure 5—figure supplement 2).

4. My main criticism is in the morphometric analysis section where the two major manipulations that might test neuronal activity's influence on glial motility suffered from: (a) being fairly underpowered with N's of 5 and 4. For a negative result that seems to be too few cells.

We have increased the N’s for analysis of motility in gabazine (n = 7 cells), TBOA (n = 7 cells), pirenzepine (n = 14 cells), and DNQX/AP5 (n = 13 cells). Due to limitations in GLAST/mTmG mouse availability in our lab currently, we performed new experiments by filling Muller glia with fluorescent dye using iontophoresis with a sharp electrode. Lateral process motility appeared similar between mGFP-expressing Muller glia and dye-loaded Muller glia in ACSF. In the presence of gabazine, TBOA, pirenzepine, or DNQX/AP5, motility appeared normal, with p-values trending further away from significance with the newly added cells. This new data has been included in modified Figure 5F and the new method for assessing motility has been added to methods (page 18 lines 403-404; page 20 lines 403-404; 445-447).

b) the blockade always came on a background of upregulation (pip on gabazine and APV/DNQX on TBOA). Even so the effects were reported as showing a non-significant trend. Is there any way to either repeat these experiments in the absence of the upregulating compounds or at least to do the statistical comparisons to the baseline in the absence of that upregulating compound. As far as I could tell baselines for these experiments compared upregulation vs upreg+ downreg which would have been an outstanding control if there had been a positive result but which is less convincing with a marginal negative result. Because the main conclusion of the entire paper largely rests on these experiments, I would prefer to see them done properly.

We have included separate experiments in which pirenzepine or DNQX/AP5 were added in the absence of gabazine or TBOA. We had originally performed motility analysis in pirenzepine or DNQX/AP5 to keep this set of experiments parallel with calcium imaging experiments in Figure 4, as well as calcium imaging experiments in a previous study from our lab which applied DNQX on a background of TBOA to block wave-associated calcium transients in Muller glia (Rosa et al., *eLife*, 2015).*Reviewer #3 (Recommendations for the authors):*

Overall this is a strong and exciting manuscript investigating important questions about neural interactions by using the retina as a model system. However, some clarifications and controls are needed to ensure that the author's main claim is fully supported by the data provided before publication.Concerns:Throughout the text, the word glia is used to depict astrocytes and Muller glia. However, the glia name also includes other cell types such as microglia and oligodendrocytes. Thus the use of glia instead of astroglial cell types may generate confusion. Particularly when generalized statements are made concerning calcium transients.

Wording has been updated where appropriate to specify when referring to Muller glia, astroglia, or glia in general.

The manuscript is well-written, clear, and accessible to a broad audience. However, the authors should improve Figures to extend the accessibility. For example, Figure 1A could start with a schematic of the IPL layers in the retina so the significance of the labeling, such as S1 S2 and so on, can be understood by diverse audiences.

Sublayers S1-S5 have been denoted on Figure 1A to set up later sublayer-specific analyses. We have made small changes to other figures as well that we hope addresses this concern.

Related to Figure 1F and G:Please clearly describe in the manuscript how the authors ensured that pharmacological treatments that stopped Muller glial process motility did not do so by damaging cell health.

We have modified the text describing results in Figure 1G to include an explanation of why we believe our cytoskeletal manipulations directly block motility, rather than indirectly via an impact on cell health (lines 104-108). Specifically, we did not observe a reactive phenotype known to occur in astroglial cells under pathological states, and Muller glial morphology appeared normal in these drugs despite the lack of motility. See Author response image 2 for an example comparing of Muller glial morphology in ACSF vs. cytochalasin-D and nocodazole. We can add this example figure to the manuscript if the reviewer/editor feels the description in the text is not sufficient.

**Author response image 2. sa2fig2:** Temporally color-coded images of 2-photon Z stack time series through the IPL of *GLAST/mTmG* retinas. Cytoskeletal manipulations using 5 µM cytochalasin-D (cyto-D) and 10 µM nocodazole (noco) blocked motility in lateral processes without inducing a morphologically reactive phenotype at P11. Scale bar 10 µm. Related to Figure 1G.

Related to figure 3B:Pirenzepine treatment affected both stalk and lateral process wave proportions. This effect is not mentioned or discussed in the Results section.

The Results section has been updated to address the reduction in lateral process responses to waves in pirenzepine and to highlight the observation that stalks are more sensitive to pirenzepine treatment than lateral processes (lines 153-156).

Is M1 mAChR only expressed by Muller glia in the retina? Citing literature investigating this question would benefit the discussion of the impacts of pharmacological manipulations on Muller Glia.

M1 mAChRs are also expressed in the retina but not in circuits that impact retinal waves. The possibility for pirenzepine to be acting on neuronal M1 mAChRs has been addressed in the modified discussion, along with literature examining mAChR expression throughout the retina (page 15, lines 324-330).

Related to figure 4:Gabazine has a strong effect on lateral process calcium event frequency, which is M1 mAChR independent. This is not mentioned in the results. But interesting.

We addressed this briefly in the Results section of our original manuscript in lines 162-173: “Gabazine also increased the proportion of stalks and lateral process ROIs that underwent wave-associated calcium transients (Figure 4A,B; Figure 4—figure supplement 1), but with differential compartment-specific effects…” This is now found in the revision with additional clarification added on page 8-9, lines 170-184. We have also included a note in the discussion about this point (page 14 lines 309-312 ).

Related to figure 5 E and F:Bath application of BAPTA was used to block all retinal waves and all calcium transients in Mueller glia. However, the evidence provided for the lack of calcium transients in lateral processes under BAPTA treatment is weak. How did the authors ensure that there were no residual calcium transients in the lateral processes of Muller glia in the presence of BAPTA? To rule out any residual calcium, a similar experiment using Lck-GCaMP in Muller glia is needed.

We have included a new supplementary figure (Figure 5—figure supplement 2) and additional clarification in the text to verify that residual calcium transients in lateral processes were eliminated in the presence of BAPTA-AM using GLAST/Lck-GCaMP6f (lines 361-365; lines 421-424).

Related to Figure 6:The authors used a genetic model to impair retinal waves and domesticated its impact on Muller glial morphology. They found no significant differences between the morphological parameters of Muller glia protrusions between the two genotypes at two different ages. While average morphological parameters seem to be similar between the genotypes, the distribution of data points looks different, indicating a role for the waves in helping Muller glia establish similar size balance territories within the synaptic layers. Comparing the distribution of data points between two genotypes can be informative.

We agree that the variance within data points changes with age for several morphological measurements presented in Figure 6, but for most measurements the variance is similar between wild type and β2-nAChR-KO retinas within each age. We have formally tested this using 2-way ANOVA, which revealed a significant main effect of age on number of branch points, process length, number of process tips, and number of branches off the stalk. ANOVA revealed a significant main effect of genotype only on the number of branches projecting off the stalk, similar to results from unpaired t-tests described in the text. We also applied F-tests for unequal variances to directly test for changes in variance between wild type and β2-nAChR-KO Muller glia morphological measures, as suggested by reviewer. However, we found no significant differences. Therefore, we do not think this analysis impacted the main conclusions. However, we can add these tables to the manuscript if the reviewer/editor feels it is important.

**Author response table 2. sa2table2:** Author response table 2. Analysis of variance (ANOVA) of morphological properties between wild type and β2-nAChR-KO Müller glia. 2-way ANOVA was used to test effects of and interaction between age and genotype on morphological properties of Muller glia in wild type (WT) and β2-nAChR-KO retinas.

Effects tested	Main: age	Main: genotype	Interaction: genotype*age
# branch points	F = 18.11p = 0.0001	F = 0.5p = 0.48	F = 94p = 0.33
stalk length (µm)	F = 0.06p = 0.81	F = 0.0p = 0.96	F = 0.0p = 0.99
process length (µm)	F = 9.61p = 0.0027	F = 0.24p = 0.63	F = 0.47p = 0.49
# tips	F = 16.69p = 0.0001	F = 0.04p = 0.84	F = 1.05p = 0.31
convex hull area (µm^2^)	F = 1.39p = 0.24	F = 1.46p = 0.23	F = 0.73p = 0.40
# branches off stalk	F = 22.08p = 1.12 x 10^-5^	F = 8.18p = 0.0055	F = 0.11p = 0.75

**Author response table 3. sa2table3:** Author response table 3. F-test for unequal variances of morphological properties between wild type and β2-nAChR-KO Müller glia. F-test for unequal variance between groups was used to compare variance of morphological properties of Muller glia at P12 and P30 between wild type and β2-nAChR-KO retinas.

Effects tested	P12 (F-statistic; p-value)	P30 (F-statistic; p-value)
# branch points	F = 0.73p = 0.44	F = 1.84p = 0.24
stalk length (µm)	F = 0.99p = 0.98	F = 3.14p = 0.06
process length (µm)	F = 0.62p = 0.23	F = 1.60p = 0.37
# tips	F = 0.89p = 0.77	F = 1.87p = 0.23
convex hull area (µm^2^)	F = 0.50p = 0.09	F = 0.39p = 0.10
# branches off stalk	F = 1.49p = 0.34	F = 1.43p = 0.49

Related to all data presented: The number of cells and mice used for each quantitative analysis should be given in the figure legend as well as in the provided statistical tables. The sex of mice used should also be included in the legends for each figure.

Related to all data presented: The number of cells and mice used for each quantitative analysis should be given in the figure legend as well as in the provided statistical tables. The sex of mice used should also be included in the legends for each figure.

All sample sizes are reported in statistical tables at the end of the manuscript. Because multiple age groups and conditions were examined for each figure, each with varying numbers of cells, lateral processes, and animals, we opted to list all sample sizes in statistical tables instead of figure legends to enable easier reading and shorter legends. Mice of both sexes were used for all experiments, as described in the Methods section (line 433).